# PUMILIO, but not RBMX, binding is required for regulation of genomic stability by noncoding RNA *NORAD*

Mahmoud M Elguindy[1,2], Florian Kopp[1], Mohammad Goodarzi[3], Frederick Rehfeld[1], Anu Thomas[1], Tsung-Cheng Chang[1], Joshua T Mendell[1,4,5,6]*

[1]Department of Molecular Biology, University of Texas Southwestern Medical Center, Dallas, United States; [2]Medical Scientist Training Program, University of Texas Southwestern Medical Center, Dallas, United States; [3]Department of Biochemistry, University of Texas Southwestern Medical Center, Dallas, United States; [4]Harold C Simmons Comprehensive Cancer Center, University of Texas Southwestern Medical Center, Dallas, United States; [5]Hamon Center for Regenerative Science and Medicine, University of Texas Southwestern Medical Center, Dallas, United States; [6]Howard Hughes Medical Institute, University of Texas Southwestern Medical Center, Dallas, United States

*For correspondence:
joshua.mendell@utsouthwestern.edu

Competing interests: The authors declare that no competing interests exist.

**Abstract** *NORAD* is a conserved long noncoding RNA (lncRNA) that is required for genome stability in mammals. *NORAD* acts as a negative regulator of PUMILIO (PUM) proteins in the cytoplasm, and we previously showed that loss of *NORAD* or PUM hyperactivity results in genome instability and premature aging in mice (Kopp et al., 2019). Recently, however, it was reported that *NORAD* regulates genome stability through an interaction with the RNA binding protein RBMX in the nucleus. Here, we addressed the contributions of *NORAD*:PUM and *NORAD*:RBMX interactions to genome maintenance by this lncRNA in human cells. Extensive RNA FISH and fractionation experiments established that *NORAD* localizes predominantly to the cytoplasm with or without DNA damage. Moreover, genetic rescue experiments demonstrated that PUM binding is required for maintenance of genomic stability by *NORAD* whereas binding of RBMX is dispensable for this function. These data provide an important foundation for further mechanistic dissection of the *NORAD*-PUMILIO axis in genome maintenance.

DOI: https://doi.org/10.7554/eLife.48625.001

## Introduction

Long noncoding RNAs (lncRNAs) have emerged as regulators of diverse biological processes. Among these transcripts, *Noncoding RNA activated by DNA damage* (*NORAD*) is particularly noteworthy, due to its unusually abundant expression in mammalian cells and tissues and strong evolutionary conservation across mammalian species. Initial studies of *NORAD* revealed that this lncRNA is required to maintain genomic stability in mammalian cells (*Lee et al., 2016*), and provided strong evidence that this function is mediated through the ability of *NORAD* to bind to and negatively regulate PUMILIO RNA binding proteins (PUM1 and PUM2) in the cytoplasm (*Lee et al., 2016*; *Tichon et al., 2016*). PUM proteins bind with high specificity to the eight nucleotide (nt) PUMILIO response element (PRE) (UGUANAUA or UGUANAUN) on target messenger RNAs (mRNAs), triggering their deadenylation, decapping, and eventual degradation (*Miller and Olivas, 2011*; *Quenault et al., 2011*). Notably, *NORAD* contains 18 conserved PREs and has the capacity to bind a large fraction of PUM1/2 within the cell, although it is not yet known whether *NORAD* limits PUM activity through a simple sequestration model or whether additional mechanisms contribute to PUM

inhibition. Regardless, loss of *NORAD* results in PUM hyperactivity and increased repression of PUM targets that include important regulators of mitosis, DNA repair, and DNA replication, resulting in a dramatic genomic instability phenotype in *NORAD*-deficient cells and mouse tissues (*Kopp et al., 2019*; *Lee et al., 2016*). Accordingly, PUM1/2 overexpression is sufficient to phenocopy loss of *NORAD* in both human cells and mice (*Kopp et al., 2019*; *Lee et al., 2016*), while PUM1/2 loss-of-function suppresses the genomic instability phenotype in *NORAD* knockout cells (*Lee et al., 2016*).

Recently, an alternative mechanism for the regulation of genomic stability by *NORAD* was proposed (*Munschauer et al., 2018*). Proteomic analysis of the *NORAD* interactome revealed an interaction with RBMX, an RNA binding protein that contributes to the DNA damage response (*Adamson et al., 2012*). Subsequent experiments suggested that the *NORAD*:RBMX interaction facilitates the assembly of a ribonucleoprotein (RNP) complex in the nucleus that includes Topoisomerase I (TOP1) and other proteins that are critical for genome maintenance. Importantly, PUM and RBMX interact with different sites on *NORAD* and function in distinct subcellular compartments. Thus, while it remains to be determined whether the *NORAD*:RBMX interaction is necessary for regulation of genomic stability, both PUM and RBMX may play important, non-mutually exclusive roles in the genome maintenance functions of *NORAD*.

Here, we further examined the mechanism by which *NORAD* functions to maintain genome stability in human cells and directly tested the requirement for PUM and RBMX interactions in this activity. RNA fluorescent in situ hybridization (FISH) using a panel of probes spanning the entire length of *NORAD*, as well as cellular fractionation studies, definitively demonstrated that this lncRNA localizes predominantly to the cytoplasm and does not detectably redistribute to the nucleus upon induction of DNA damage. Genetic rescue experiments in *NORAD* knockdown cells established that PUM binding is essential for maintenance of genomic stability whereas interaction with RBMX is completely dispensable for this function. Further experiments demonstrated that RBMX is not required for induction of *NORAD* following DNA damage nor its cytoplasmic localization. Together, these studies establish the importance of the *NORAD*:PUM axis in regulating genomic stability in mammalian cells and provide a foundation for further dissection of the mechanism and physiologic role of this pathway.

## Results and discussion

### *NORAD* localizes predominantly to the cytoplasm with or without DNA damage

Initial studies of *NORAD* reported a predominantly cytoplasmic localization of this lncRNA in human cell lines, based on RNA FISH using pools of fluorescently-labeled oligonucleotide probes and subcellular fractionation experiments (*Lee et al., 2016*; *Tichon et al., 2016*). Recently, however, a distinct localization pattern was reported based upon RNA FISH performed using a commercially-available kit with a proprietary set of oligonucleotide probes that hybridize to an unknown segment of *NORAD* (*Munschauer et al., 2018*). In these more recent experiments, *NORAD* was reported to localize equally between the nucleus and cytoplasm and appeared to redistribute almost entirely to the nuclear compartment upon treatment of cells with the DNA damaging agents camptothecin and doxorubicin. Importantly, a single cell line (human colon cancer cell line HCT116) was used in both the previous (*Lee et al., 2016*) and more recent studies (*Munschauer et al., 2018*), arguing against a cell-type specific difference in *NORAD* trafficking as the cause of these discordant results.

We considered the possibility that the disparate observed localization patterns could be due to unrecognized processing of the *NORAD* transcript, such that different segments of the RNA that are recognized by the different FISH probes accumulate in distinct subcellular compartments. To investigate this possibility, and to reliably establish the localization of the entire *NORAD* transcript, we generated a panel of 11 in vitro transcribed RNA FISH probes spanning the complete *NORAD* sequence (*Figure 1A*) (*Mito et al., 2016*). One probe, that recognized a segment of *NORAD* containing an Alu repeat element (probe 7), gave rise to a nonspecific nuclear signal that was present in both *NORAD*$^{+/+}$ and *NORAD*$^{-/-}$ HCT116 cells. The remaining 10 probes produced a highly consistent, predominantly cytoplasmic, punctate localization pattern in wild-type cells that was absent in

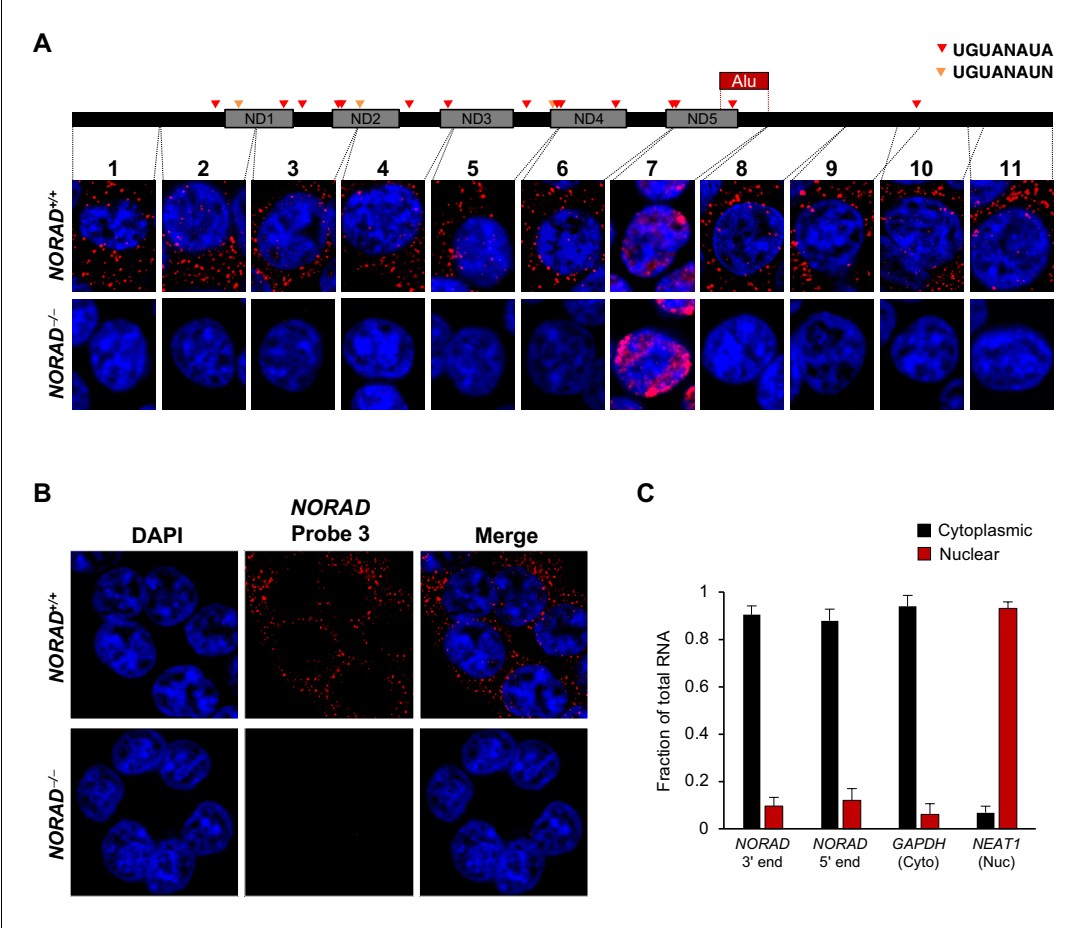

**Figure 1.** *NORAD* localizes predominantly to the cytoplasm. (**A**) RNA FISH in HCT116 cells using a panel of 11 probes tiling the entire *NORAD* transcript reveals a predominantly cytoplasmic signal that is absent in *NORAD*[−/−] cells with all probes except probe 7, which produces a nonspecific signal likely due to the presence of an Alu repeat element. *NORAD* FISH signal in red, DAPI counterstain in blue. Locations of PREs indicated by arrowheads. ND1-ND5 represent repetitive *NORAD* domains, as previously described (*Lee et al., 2016*). (**B**) RNA FISH image using probe 3 showing a wider field of cells. (**C**) Subcellular fractionation followed by qRT-PCR in HCT116 cells using primers located at the 3′ or 5′ end of *NORAD*, in *GAPDH* (cytoplasmic control), or in *NEAT1* (nuclear control). n = 3 biological replicates each with three technical replicates.
DOI: https://doi.org/10.7554/eLife.48625.002

*NORAD*[−/−] cells (*Figure 1A–B*). These results were confirmed using subcellular fractionation followed by quantitative reverse transcriptase-PCR (qRT-PCR) using primers located at the 3′ or 5′ ends of *NORAD*, which revealed that 80–90% of the transcript is localized to the cytoplasmic compartment (*Figure 1C*).

Next, we examined *NORAD* localization following treatment of cells with agents that induce DNA damage (*Figure 2—figure supplement 1A–B*). RNA FISH using the panel of probes spanning *NORAD* revealed clear cytoplasmic localization after treatment with doxorubicin or camptothecin, without a significant increase in nuclear signal compared to untreated cells (*Figure 2A–B*). Consistent with the previously reported induction of *NORAD* after DNA damage (*Lee et al., 2016*), a clear increase in cytoplasmic *NORAD* signal was apparent in treated cells (*Figure 2B*). These findings were further corroborated by subcellular fractionation experiments following treatment with DNA damaging agents, which confirmed that *NORAD* remained predominantly in the cytoplasmic compartment at all time points (*Figure 2C* and *Figure 2—figure supplement 1C*). Interestingly, we observed a modest increase in nuclear *NORAD* levels that peaked after 12 hr of camptothecin or doxorubicin treatment. We speculated that this might represent a burst of *NORAD* transcription in response to accumulating DNA damage. To test this hypothesis, we co-treated cells with DNA damaging agents and the transcriptional inhibitor actinomycin D. As expected, this completely

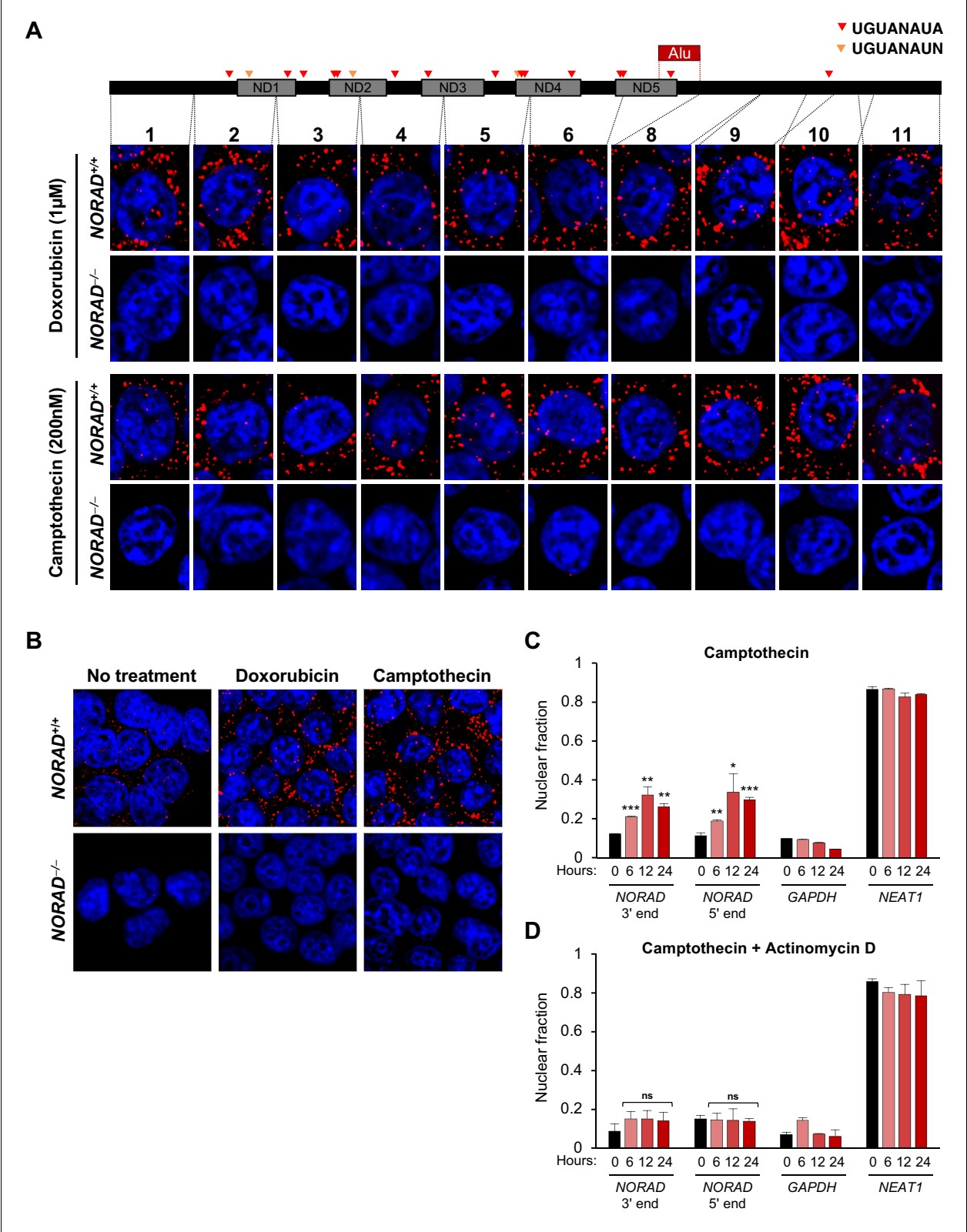

**Figure 2.** *NORAD* remains predominantly in the cytoplasm after treatment with DNA damaging agents. (**A**) RNA FISH in HCT116 cells using the indicated *NORAD* probes following a 12 hr treatment with doxorubicin or camptothecin. (**B**) *NORAD* RNA FISH (probe 5) after the indicated drug treatments. Images captured with identical microscope settings. (**C–D**) Subcellular fractionation followed by qRT-PCR in HCT116 cells after treatment

*Figure 2 continued on next page*

eLIFE Research advance

Chromosomes and Gene Expression | Genetics and Genomics

*Figure 2 continued*

with camptothecin (C) or camptothecin plus actinomycin D (D) for the indicated number of hours. n = 3 biological replicates each with three technical replicates. ns, not significant; *p<0.05; **p<0.01; ***p<0.001; one-tailed t-test comparing each sample to the 0 hr time-point.

DOI: https://doi.org/10.7554/eLife.48625.003

The following figure supplement is available for figure 2:

**Figure supplement 1.** *NORAD* remains predominantly cytoplasmic following doxorubicin-induced DNA damage.

DOI: https://doi.org/10.7554/eLife.48625.004

abrogated any detectable increase in nuclear *NORAD* abundance in treated cells (*Figure 2D* and *Figure 2—figure supplement 1D*).

These comprehensive RNA FISH and subcellular fractionation experiments provide definitive evidence that *NORAD* is a predominantly cytoplasmic RNA in HCT116 cells and does not detectably redistribute to the nucleus upon DNA damage. These findings are consistent with the reported localization of *NORAD* in other human and mouse cell lines (*Kopp et al., 2019*; *Lee et al., 2016*; *Tichon et al., 2016*). We speculate that the disparate localization pattern observed using a commercially-available RNA FISH probe set (*Munschauer et al., 2018*) most likely represented a non-specific signal.

## PUM1, PUM2, and RBMX are components of the *NORAD* interactome

Previous crosslinking-immunoprecipitation coupled with high throughput sequencing (CLIP-seq) studies demonstrated that *NORAD* is the preferred binding partner of PUM2 in both human cells (*Lee et al., 2016*) and mouse brain (*Kopp et al., 2019*). In light of these findings, it was surprising that PUM1/2 were not reported among the most enriched *NORAD*-bound proteins in the recent RNA antisense purification with quantitative mass spectrometry (RAP-MS) experiments performed in HCT116 cells that identified the *NORAD*:RBMX interaction (*Munschauer et al., 2018*). Since these RAP-MS experiments utilized pulse labeling with 4-thiouridine to crosslink *NORAD* to protein interactors, a bias towards detection of proteins that bind to newly synthesized *NORAD* would be expected, potentially explaining the enrichment of nuclear interactors observed. Nevertheless, we reanalyzed the published RAP-MS dataset to determine whether PUM1 or PUM2 were enriched in *NORAD* pull-downs compared to control *RMRP* pull-downs. Peptides were identified and scored using a combined algorithm that employed three search engines (Sequest HT, Mascot, and MS Amanda). Isoforms of PUM1 and PUM2, along with RBMX, were indeed identified as significantly-enriched interacting partners of *NORAD* compared to *RMRP* (*Figure 3—figure supplement 1*). Notably, PUM1 was more enriched than PUM2 in our analysis, which may reflect its higher abundance in HCT116 cells (*Lee et al., 2016*). These results confirmed that both PUM proteins and RBMX are identified by RAP-MS as significant *NORAD*-interacting partners.

## Binding of PUMILIO, but not RBMX, to *NORAD* is necessary for genome stability

Genetic epistasis experiments have strongly implicated a role for PUM1/2 in the regulation of genomic stability by *NORAD*, with PUM2 overexpression phenocopying, and PUM1/2 knockdown suppressing, the effects of *NORAD* deletion (*Kopp et al., 2019*; *Lee et al., 2016*). Nevertheless, it has not yet been directly tested whether binding of PUM1/2 is required for *NORAD* function. Similarly, a requirement for RBMX binding in genome maintenance by *NORAD* has not been established. Therefore, to directly interrogate the importance of PUM and RBMX binding for *NORAD* function, we generated a series of mutant *NORAD* constructs lacking either PUM or RBMX binding sites (*Figure 3A*). For each of the 18 PREs within *NORAD*, the UGU sequence, which is essential for PUM binding (*Bohn et al., 2018*; *Van Etten et al., 2012*), was mutated to ACA to abolish PUM interaction (PREmut construct). To remove the RBMX binding site, the first 898 nucleotides (nt) of *NORAD* were deleted (*Munschauer et al., 2018*) (5′ deletion construct). We also sought to determine whether PUM or RBMX binding regions of *NORAD* could represent minimal functional domains that are sufficient for maintaining genomic stability. To this end, we generated a fragment comprising *NORAD* domain 4 (ND4), which represents the most conserved of five repeated segments within this lncRNA termed *NORAD* domains (*Lee et al., 2016*) and contains 4 PREs (nt 2494–3156). An RBMX binding

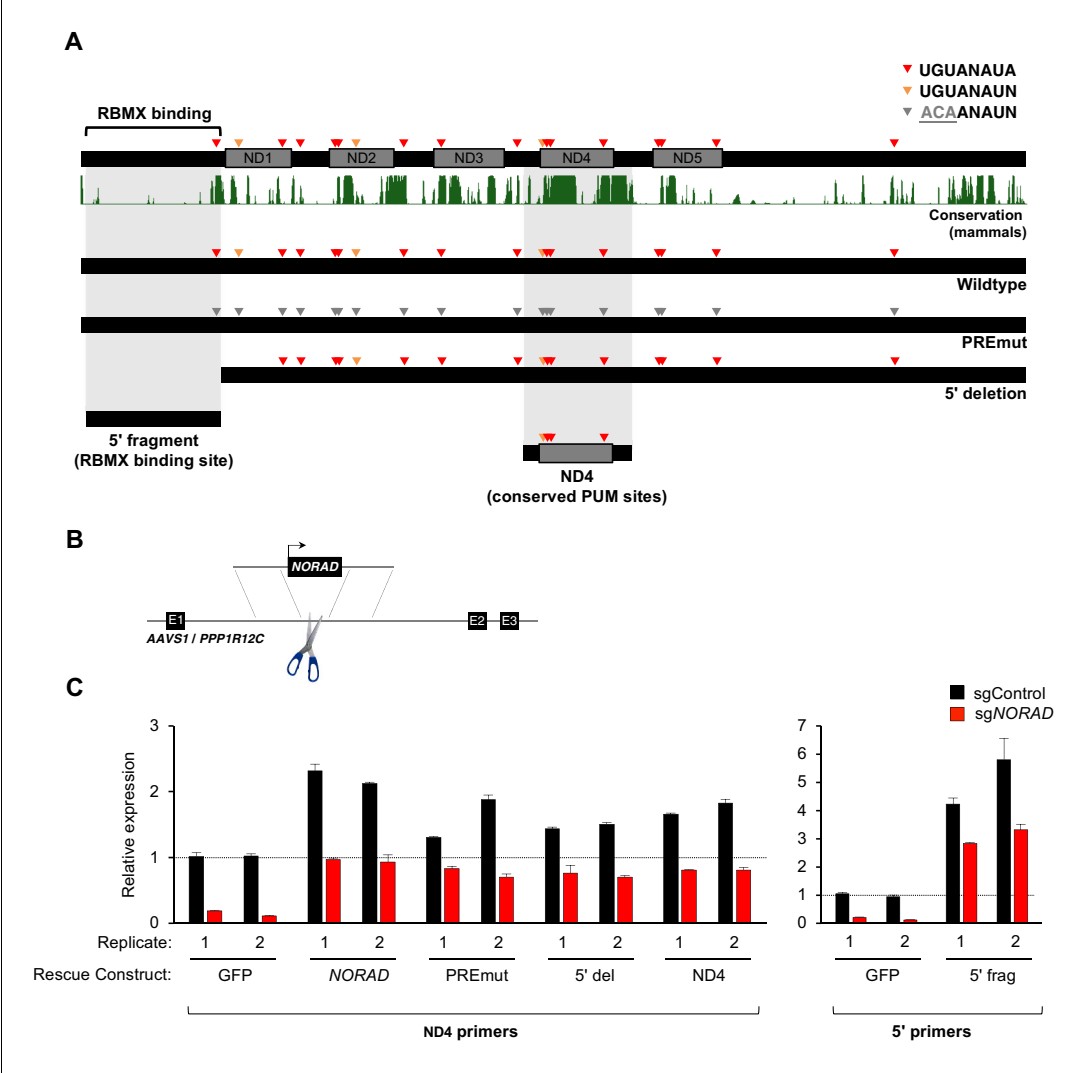

**Figure 3.** Generation and stable expression of *NORAD* constructs. (**A**) Schematic depicting wild-type or mutant *NORAD* constructs. *NORAD* sequence conservation in mammals (UCSC Genome Browser Hg38 PhastCons track) highlights the strong conservation of the region of *NORAD* harboring PREs (arrowheads). PREmut contains 18 UGU to ACA mutations in PREs (gray arrowheads); 5′ deletion (5′ del) lacks the RBMX binding site (nt 1–898) (***Munschauer et al., 2018***); 5′ fragment (5′ frag) spans the RBMX binding site (nt 33–898); ND4 construct represents the most conserved segment of *NORAD* (nt 2494–3156). (**B**) Schematic depicting insertion of constructs into the *AAVS1/PPP1R12C* locus using TALENs. (**C**) qRT-PCR analysis of expression of each *NORAD* construct in HCT116 CRISPRi cells after infection with control or endogenous *NORAD*-targeting sgRNAs. Expression was normalized to endogenous *NORAD* level, represented by expression in *AAVS1*-GFP cells infected with sgControl (replicate 1 samples normalized to sgControl *AAVS1*-GFP replicate 1; replicate 2 samples normalized to sgControl *AAVS1*-GFP replicate 2). The data in the left graph were generated with a primer pair in ND4 that does not amplify the 5′ fragment, while the right graph used primers at the *NORAD* 5′ end. Replicates represent two independently-derived *AAVS1* knock-in and sgRNA-infected cell lines. Values normalized to *GAPDH* expression. n = 3 technical replicates per sample.
DOI: https://doi.org/10.7554/eLife.48625.005

The following figure supplement is available for figure 3:

**Figure supplement 1.** Reanalysis of *NORAD* RAP-MS data.
DOI: https://doi.org/10.7554/eLife.48625.006

site fragment, representing the 5′ end of *NORAD* (nt 33–898), which also harbors one PRE, was also generated (5′ fragment).

Wild-type or mutant *NORAD* constructs, as well as a control GFP sequence, under the control of a constitutive promoter were introduced into the *AAVS1/PPP1R12C* locus of HCT116 cells using a previously published TALEN pair (***Sanjana et al., 2012***) (***Figure 3B***). Endogenous *NORAD* was then depleted using CRISPR interference (CRISPRi) with a single-guide RNA (sgRNA) targeting the

endogenous *NORAD* promoter. As expected, in cells infected with a non-targeting control sgRNA (sgControl), increased total levels of *NORAD* were observed upon expression of *NORAD* rescue constructs in trans (~1.5–6 fold overexpression, depending on the construct) (*Figure 3C*). Upon silencing of the endogenous *NORAD* locus, however, near physiologic expression levels for all constructs were achieved, with the exception of the 5′ fragment which exhibited ~3 fold overexpression, perhaps indicating increased stability of this transcript segment when expressed in isolation.

We next used UV crosslinking and RNA immunoprecipitation (RIP) to assess binding of wild-type and mutant *NORAD* transcripts to endogenous PUM1, PUM2, and RBMX. Pull-downs of each of these proteins resulted in the expected enrichment of wild-type *NORAD*, but not *GAPDH*, relative to immunoprecipitation with control IgG (*Figure 4A* and *Figure 4—figure supplement 1*). The PRE-mut transcript as well as the 5′ fragment did not bind to PUM1/2 but were recovered in RBMX RIP samples as efficiently as wild-type *NORAD*. In contrast, the 5′ deletion construct and ND4 fragment retained PUM1 and PUM2 binding activity, but interaction with RBMX was not detectable above background. Furthermore, RNA FISH documented a predominantly cytoplasmic localization pattern of each construct (*Figure 4B*).

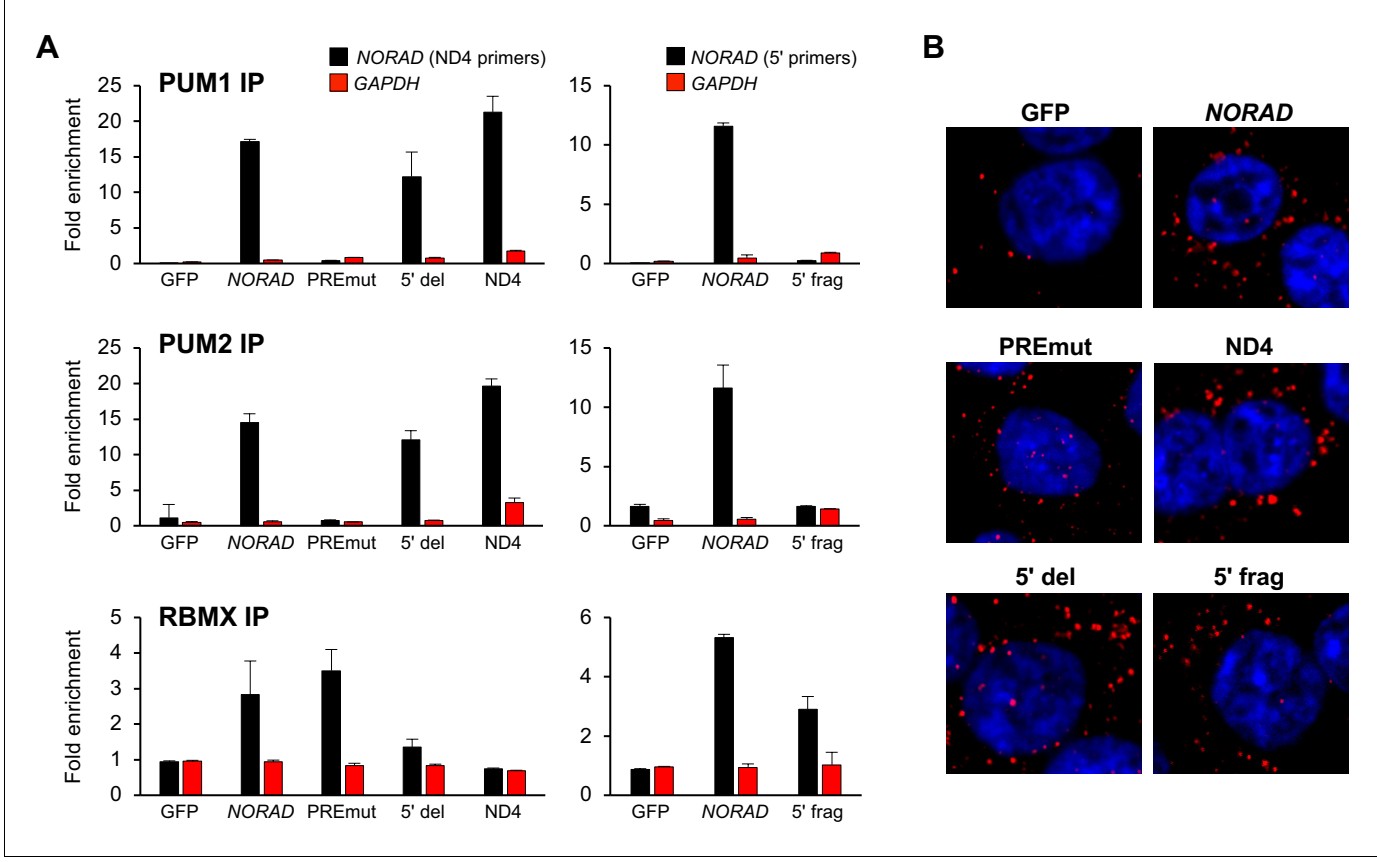

**Figure 4.** RNA immunoprecipitation and localization of *NORAD* constructs. (**A**) UV crosslinking and RNA immunoprecipitation (RIP) was used to assess PUM1, PUM2, and RBMX interactions with GFP mRNA or the indicated *NORAD* constructs. After knock-in of the indicated constructs to the *AAVS1* locus in HCT116 CRISPRi cells, endogenous *NORAD* was silenced with a lentivirally-expressed sgRNA. qRT-PCR was used to assess *NORAD* or *GAPDH* recovery in each RIP sample, expressed as fold-enrichment over pull-down with IgG. The data in the left graphs were generated with a primer pair in ND4 that does not amplify the 5′ fragment, while the right graphs used primers at the *NORAD* 5′ end. n = 2 biological replicates, each measured with three technical replicates. (**B**) Representative RNA FISH images of wild-type or mutant *NORAD* transcripts expressed from the *AAVS1* locus in HCT116 CRISPRi cells after knockdown of endogenous *NORAD*. Probe 10 was used for full-length *NORAD*, PREmut, and 5′ del constructs; probe 1 was used for 5′ frag; and probe 6 was used for ND4.

DOI: https://doi.org/10.7554/eLife.48625.007

The following figure supplement is available for figure 4:

**Figure supplement 1.** Representative western blots of PUM1, PUM2, and RBMX in RIP experiments.

DOI: https://doi.org/10.7554/eLife.48625.008

Genome stability was first assessed in cell populations expressing wild-type or mutant *NORAD* constructs by quantifying the number of aneuploid cells in each population using DNA FISH for marker chromosomes 7 and 20, as described previously (*Lee et al., 2016*) (*Figure 5A–B*). Importantly, no significant increase in aneuploidy was observed in any of the *NORAD* rescue populations after infection with non-target sgRNA (sgControl), indicating that overexpression of the *NORAD* constructs in trans did not trigger genomic instability. Furthermore, as expected, knockdown of endogenous *NORAD* in GFP-control cells resulted in a significant accumulation of aneuploid cells. The frequency of aneuploidy observed under these conditions was very similar to that observed previously in *NORAD*$^{-/-}$ HCT116 cells (*Lee et al., 2016*). Expression of wild-type *NORAD* in trans was sufficient to fully suppress the accumulation of aneuploid cells after silencing endogenous *NORAD*. Cells expressing the PREmut transcript, however, exhibited high levels of aneuploidy, demonstrating that loss of PUM binding abrogated the ability of *NORAD* to maintain genomic stability. Moreover, the 5′ deletion construct that lacks the RBMX binding site, but preserves the PUM interaction, was fully functional in this assay and completely prevented the accumulation of aneuploid cells. Thus, RBMX binding to *NORAD* is dispensable for genome maintenance. Remarkably, we observed a strong suppression of aneuploidy in cells expressing the minimal ND4 fragment, further supporting the centrality of the PUM interaction for *NORAD* function, while the 5′ fragment of *NORAD* had no activity in this assay.

To further assess the role of RBMX and PUM binding in genome maintenance by *NORAD*, we quantified the frequency of chromosomal segregation defects in HCT116 cell populations expressing the various *NORAD* rescue constructs. We have previously demonstrated that *NORAD* knockout cells exhibit a significant increase in mitotic errors (*Lee et al., 2016*) and the same phenotypic assay was later used by Munschauer et al. to confirm genomic instability in *NORAD* and *RBMX* knockdown cells (*Munschauer et al., 2018*). Examination of DAPI-stained anaphase nuclei revealed the expected increase in chromosomal segregation defects in GFP-expressing control cells following *NORAD* knockdown (*Figure 5C–D*). Consistent with our analyses of aneuploidy using DNA FISH, expression of full-length *NORAD*, or the 5′ deletion construct lacking the RBMX binding site, reversed this phenotype and suppressed the increase in mitotic errors. We further documented that mutation of the PUM binding sites (PREmut) abolished *NORAD* function in this assay. Finally, we found that the ND4 segment containing several highly conserved PUM binding sites, but not the 5′ fragment encompassing the RBMX binding site, exhibited significant rescue activity in this assay. Overall, these data provide compelling evidence that PUM, but not RBMX, binding to *NORAD* is necessary for the maintenance of genomic stability by this lncRNA.

## RBMX is not required for *NORAD* expression or localization

Although RBMX is not required for maintenance of genomic stability by *NORAD*, we were able to confirm binding of this protein to the 5′ end of *NORAD*, as reported (*Munschauer et al., 2018*) (*Figure 3* and *Figure 4A*). Thus, we investigated whether RBMX functions as an upstream regulator of *NORAD* expression or localization. Depletion of RBMX using CRISPRi resulted in an increase in *NORAD* expression that was further augmented by doxorubicin treatment (*Figure 6A*). We speculate that this increase in *NORAD* levels may be an indirect effect of the previously reported accumulation of DNA damage caused by RBMX loss of function (*Adamson et al., 2012*). Additionally, RBMX knockdown did not alter the predominantly cytoplasmic localization of *NORAD*, as indicated by subcellular fractionation experiments (*Figure 6B*). We conclude that RBMX is not an essential cofactor for *NORAD* expression or localization.

In sum, these results establish the essential role of PUM binding for the regulation of genomic stability by *NORAD*. A systematic examination of the subcellular localization of this lncRNA unequivocally established its predominantly cytoplasmic localization under baseline conditions as well as after treatment with DNA-damaging agents. Moreover, genetic complementation experiments demonstrated that PUM binding is essential, whereas RBMX interaction is dispensable, for the genome maintenance function of *NORAD*. These results further define and clarify the *NORAD* molecular mechanism of action and direct future investigation towards elucidation of the regulation and physiologic roles of the *NORAD*:PUM axis.

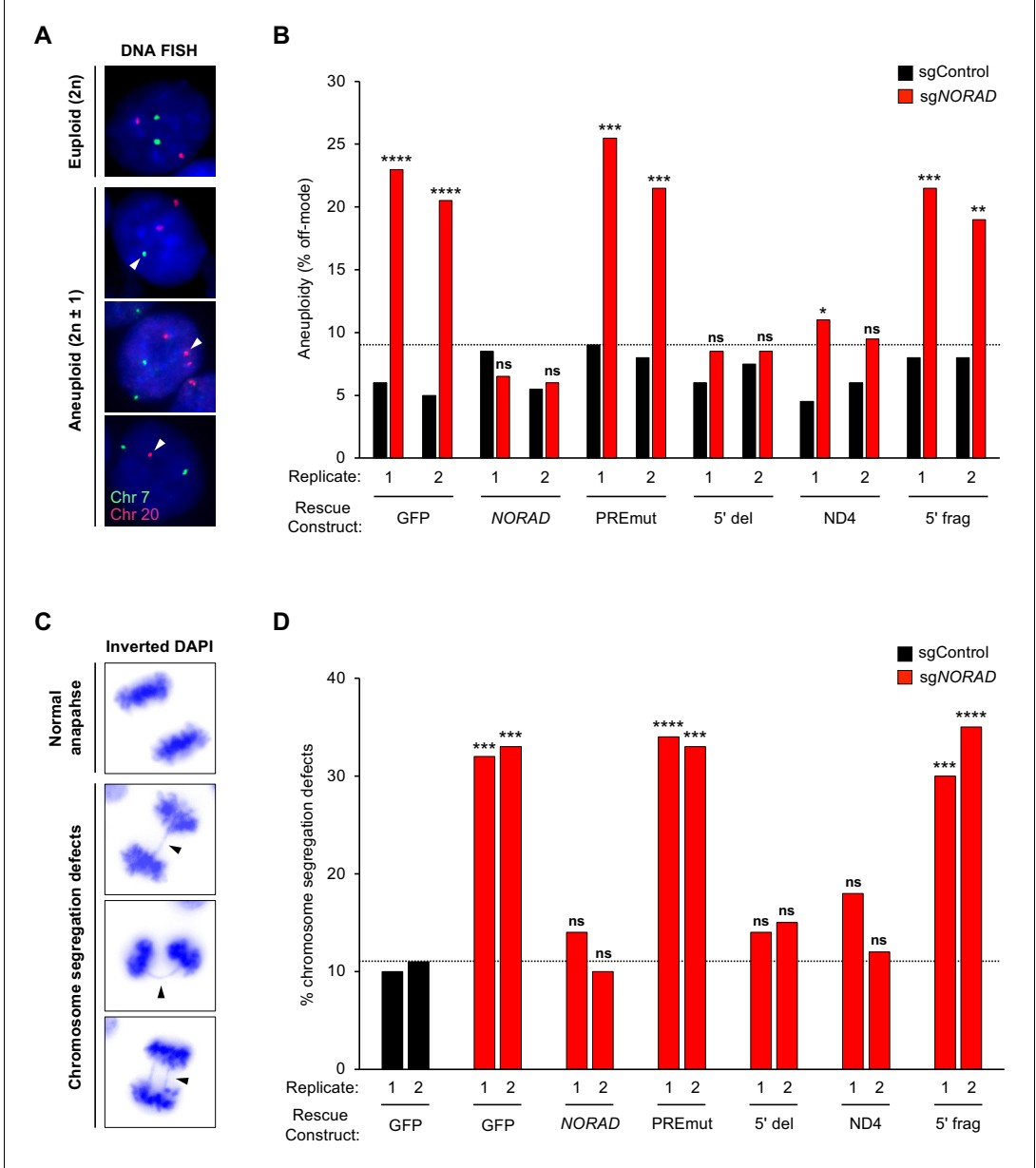

**Figure 5.** PUMILIO, but not RBMX, binding to *NORAD* is necessary for genome stability. (**A**) Representative DNA FISH images for chromosome 7 (green) and 20 (red) showing examples of cells with modal (2 n) and non-modal (2n ± 1) chromosome numbers. Arrowheads indicate chromosome gain or loss. (**B**) HCT116 CRISPRi cells stably expressing the indicated *AAVS1* knock-in construct were infected with lentivirus expressing control or endogenous *NORAD*-targeting sgRNA. Aneuploidy was assayed 18–21 days later using DNA FISH for chromosome 7 and 20, and the frequency of interphase cells exhibiting a non-modal (2 n) chromosome number was scored. Replicates represent two independently-derived *AAVS1* knock-in and sgRNA-infected cell lines. 200 nuclei were scored per sample. The dotted line denotes the highest level of background aneuploidy observed in sgControl-infected cells. ns, not significant; *p<0.05; **p<0.01; ***p<0.001; ****p<0.0001, chi-square test comparing sg*NORAD* to sgControl for each replicate. (**C**) Representative images of anaphase cells with normal or abnormal (arrowheads) chromosome segregation in DAPI-stained HCT116 CRISPRi cells. (**D**) The frequency of mitotic cells exhibiting chromosome segregation defects was determined in both biological replicates of each cell population (100 anaphase cells assayed per sample). The dotted line denotes the highest percentage of chromosome segregation defects observed in sgControl-infected cells. ns, not significant; *p<0.05; **p<0.01; ***p<0.001; ****p<0.0001, chi-square test comparing all sg*NORAD* replicate 1 samples to sgControl GFP replicate 1 and all sg*NORAD* replicate 2 samples to sgControl GFP replicate 2.
DOI: https://doi.org/10.7554/eLife.48625.009

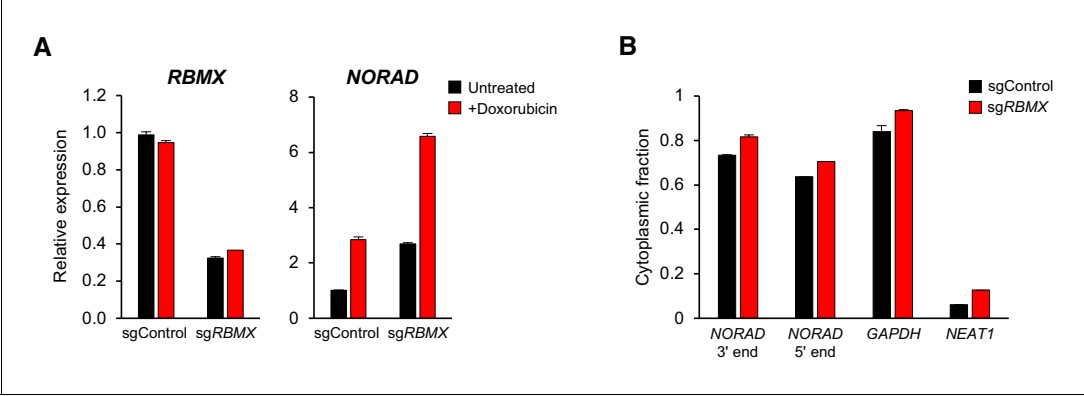

**Figure 6.** RBMX is not required for *NORAD* expression or localization. (**A**) qRT-PCR analysis of *RBMX* and *NORAD* transcript levels in HCT116 CRISPRi cells after introduction of the indicated lentivirally-expressed sgRNA with or without doxorubicin treatment (1 µM for 24 hr). Quantification relative to *GAPDH*. n = 3 technical replicates. (**B**) Subcellular fractionation and qRT-PCR of *NORAD*, *GAPDH* (cytoplasmic control), or *NEAT1* (nuclear control) following introduction of control or *RBMX*-targeting sgRNAs. n = 3 biological replicates each with three technical replicates.
DOI: https://doi.org/10.7554/eLife.48625.010

# Materials and methods

## Key resources table

| Reagent type (species) or resource | Designation | Source or reference | Identifiers | Additional information |
|---|---|---|---|---|
| Gene (*Homo sapiens*) | *NORAD* (*LINC00657*) | NA | Ensembl: ENSG00000260032 | |
| Cell line (*Homo sapiens*) | HCT116 | ATCC | CCL-247, RRID:CVCL_0291 | |
| Cell line (*Homo sapiens*) | *NORAD*<sup>-/-</sup> HCT116 | *Lee et al., 2016* | | |
| Cell line (*Homo sapiens*) | HCT116 CRISPRi | this paper; see Materials and methods section Cell culture and generation of HCT116 CRISPRi cell line | | |
| Antibody | Anti-PUM1 (polyclonal rabbit) | Santa Cruz | sc-135049, RRID:AB_10610604 | RIP |
| Antibody | Anti-PUM2 (polyclonal goat) | Santa Cruz | sc-31535, RRID:AB_654939 | RIP |
| Antibody | Anti-RBMX (monoclonal rabbit) | Cell Signaling | #14794, RRID:AB_2798614 | RIP, WB (1:1000) |
| Antibody | Anti-PUM2 (monoclonal rabbit) | Abcam | ab92390, RRID:AB_10563318 | WB (1:1000) |
| Antibody | Anti-PUM1 (monoclonal rabbit) | Abcam | ab92545, RRID:AB_10563695 | WB (1:1000) |
| Antibody | Anti-GAPDH (monoclonal rabbit) | Cell Signaling | #2118, RRID:AB_561053 | WB (1:5000) |
| Antibody | Anti-phospho-histone H2A.X (Ser139) | Cell Signaling | #2577, RRID:AB_2118010 | IF (1:500), WB (1:1000) |
| Antibody | IRDye 800CW anti-rabbit (donkey) | Licor | 925–32213, RRID:AB_2715510 | WB (1:10000) |
| Antibody | Anti-Digoxigenin (monoclonal mouse) | Roche | 11333062910, RRID:AB_514495 | RNA FISH |

*Continued on next page*

*Continued*

| Reagent type (species) or resource | Designation | Source or reference | Identifiers | Additional information |
|---|---|---|---|---|
| Antibody | Anti-Mouse IgG, Cy3 (polyclonal goat) | EMD Millipore | AP124C | RNA FISH |
| Recombinant DNA reagent | *AAVS1/PPP1R12C* targeting vector | Addgene | #22072, RRID:Addgene_22072 | |
| Recombinant DNA reagent | hAAVS1 1L TALEN; hAAVS1 1R TALEN | Addgene | #35431, RRID:Addgene_35431; #35432, RRID:Addgene_35432 | |
| Recombinant DNA reagent | pU6-sgRNA EF1a -PuroR-T2A-BFP | Addgene | #60955, RRID:Addgene_60955 | CRISPRi-mediated knockdown |
| Sequence-based reagent | Chromosome enumeration Probe Chr. 7 (green) | Empire Genomics | CHR07-10-GR | DNA FISH |
| Sequence-based reagent | Chromosome enumeration Probe Chr. 20 (red) | Empire Genomics | CHR20-10-RE | DNA FISH |
| Software, algorithm | Prism 7 | GraphPad Software | | |
| Software, algorithm | Proteome Discoverer | Thermo Fisher | | |
| Software, algorithm | Limma package for R | *Smyth, 2004* | | |

## Cell culture and generation of HCT116 CRISPRi cell line

HCT116 cells (ATCC) were cultured in McCoy's 5a media (Thermo Fisher Scientific) supplemented with 10% FBS (Gibco, Sigma-Aldrich) and 1X AA (Gibco). The cell line was authenticated by ATCC using short tandem repeat (STR) analysis in November 2017. All cell lines were confirmed to be free of mycoplasma contamination. HCT116 *NORAD*$^{-/-}$ cells were generated previously (*Lee et al., 2016*).

To generate the HCT116 CRISPRi cell line, lentivirus expressing dCas9/BFP/KRAB was produced by first seeding $6 \times 10^5$ HEK293T cells per well in a six-well plate. The following day, cells were transfected with 1.4 µg of pHR-SFFV-dCas9-BFP-KRAB (Addgene plasmid #46911), 0.84 µg of psPAX2 (Addgene plasmid #12260), 0.56 µg of pMD2.G (Addgene plasmid #12259), 8.4 µl of FuGENE HD (Promega), and 165 µl Opti-MEM (Thermo Fisher) according to the manufacturer's instructions. Medium was changed the next day. Two days after transfection, medium was collected and passed through a 0.45 µm SFCA sterile filter. Recipient HCT116 cells were transduced overnight using medium supplemented with 8 µg/ml polybrene (EMD Millipore). Cells expressing BFP were enriched by FACS and single-cell clonal lines were derived.

## RNA Fluorescent in situ hybridization (RNA FISH)

RNA FISH was performed as described previously (*Mito et al., 2016*) with the following modifications. DIG-labeled RNA probes for human *NORAD* were synthesized by in vitro transcription using a DIG-labeling mix (Roche). Primers used for amplification of the DNA template for each probe are provided in *Supplementary file 1*. $2 \times 10^5$ cells were grown on poly-L-lysine coated coverslips for 24 to 36 hr. For RNA FISH experiments with DNA damage treatment, cells were grown for 24 hr and treated with either 1 µM doxorubicin or 200 nM camptothecin for an additional 12 hr. Samples were rinsed twice in phosphate buffered saline (PBS), fixed in 4% paraformaldehyde for 10 min, washed again in PBS, and permeabilized in 0.5% Triton X-100 for 10 min. Samples were then washed twice with PBS and rinsed with DEPC-treated water prior to incubation in prehybridization buffer (50% formamide, 2X SSC, 1X Denhardt's solution, 10 mM EDTA, 0.1 mg/ml yeast tRNA, 0.01% Tween-20) for 1 hr. 10 ng/µl DIG-labeled RNA probe was diluted in hybridization buffer (prehybridization buffer with 5% dextran sulfate) and used for hybridization at 55°C for 16 to 20 hr. Following hybridization, samples were washed, treated with RNase A, and blocked using Blocking Reagent (Roche). DIG-

labeled probes were detected using mouse monoclonal anti-DIG primary antibody (Roche; 1:100 dilution) and a Cy3-labeled goat anti-mouse IgG secondary antibody (Roche; 1:100 dilution). Immunofluorescence and western blot analysis of the DNA damage marker γ-H2AX was performed using anti-γ-H2AX (Ser139) antibody (2577, Cell Signaling). Samples were mounted using SlowFade Diamond Antifade with DAPI mounting media (Invitrogen) and imaging was performed using a Zeiss LSM700 confocal microscope. ImageJ was used for further image analysis.

## Subcellular fractionation

Cells were seeded in triplicate and $1 \times 10^6$ cells were collected for subcellular fractionation, which was performed as previously described (*Kopp et al., 2019*; *Lee et al., 2016*). Briefly, cell pellets were lysed in RLN1 buffer (50 mM Tris-HCl pH 8.0, 140 mM NaCl, 1.5 mM MgCl$_2$, 0.5% NP-40, RNAse inhibitor) on ice for 5 min and centrifuged at 500g × 2 min. The supernatant containing the cytoplasmic fraction was separated from the pelleted nuclear fraction. RNA was then isolated from both fractions using the Qiagen RNeasy kit and equal cell equivalents of nuclear and cytoplasmic RNA were used in subsequent qRT-PCR reactions. All samples were tested for *NORAD* as well as *NEAT1* (nuclear control) and *GAPDH* (cytoplasmic control). The sum of the nuclear and cytoplasmic expression level of each transcript was set to 100%, and the percentage of each transcript localized to each compartment was determined. *NEAT1* and *GAPDH*, respectively, showed the expected nuclear and cytoplasmic localization in each experiment, confirming successful fractionation.

## Reanalysis of *NORAD* RAP-MS data

The raw mass spectra files from iTRAQ-labeled *NORAD* and *RMRP* RAP-MS experiments (*Munschauer et al., 2018*) were downloaded from MassIVE (https://massive.ucsd.edu) using the identifier: MSV000082561. Peptide identification and quantification was performed using Proteome Discoverer (Thermo Fisher) with three search engines combined (Sequest HT, Mascot, and MS Amanda). MS/MS spectra were searched against the human Uniprot database. Search parameters included: trypsin enzyme specificity with a maximum of 2 missed cleavages tolerated, False Discovery Rate (FDR) set to 0.01 at both peptide and protein level, ±10 ppm for precursor mass tolerance with a shorter window for fragment mass tolerance for the first search, and carbamidomethylation of cysteine modification and iTRAQ labels on N-termini and lysine residues as fixed modifications and oxidation of methionine and N-termini acetylation as variable modifications. All peptide and protein identifications had scores surpassing the combined search engine significance threshold for identification. Protein abundance was calculated as the intensity given from precursor quantification and was then normalized to the total peptide amount. To correct for total abundance differences between samples, protein and peptide abundance values in each sample were corrected by a constant factor such that the end total abundance was equivalent across all samples. Fold change was calculated as the log$_2$ difference of average scaled protein abundance in *NORAD* samples and *RMRP* sample. For statistical analysis, we used the limma package (*Smyth, 2004*) in R (https://www.r-project.org/) to calculate the adjusted p-value using a moderated *t*-test and Benjamini Hochberg method to control the FDR.

## RNA isolation and quantitative reverse transcription PCR (qRT-PCR)

RNA was isolated from cells using the RNeasy Mini Kit (Qiagen), or, for RIP experiments, Trizol (Invitrogen), and treated with RNase-free DNase (Qiagen). RNA was reverse transcribed with PrimeScript RT-PCR mix (Clonetech), and Power SYBR Green PCR Master Mix (Applied Biosystems) was used for qPCR. Biological replicates represent independently grown and processed cells. Technical replicates represent multiple measurements of the same biological sample. Primer sequences are provided in *Supplementary file 1*.

## Generation and *AAVS1* knock-in of *NORAD* constructs

Full-length wild-type *NORAD* was amplified from a modified pcDNA3.1 vector containing the *NORAD* cDNA (*Lee et al., 2016*), along with an additional 115 base pairs downstream of the endogenous *NORAD* polyadenylation site. The PRE-mutant (PREmut) construct containing 18 PRE mutations (TGT to ACA) was synthesized by GENEWIZ. The 5′ deletion construct (Δ1–898), 5′ fragment (nt 33–898), and ND4 were amplified from the full-length *NORAD* construct using primers provided

in *Supplementary file 1*. Constructs were cloned into a *AAVS1/PPP1R12C* targeting vector (*AAVS1* hPGK-PuroR-pA donor, Addgene plasmid #22072) modified by replacing the puromycin resistance gene with a hygromycin resistance gene and digested with KpnI and MfeI to remove the GFP cassette. These vectors, as well as a control GFP vector, were then inserted into the *AAVS1* locus of HCT116 CRISPRi cells using a previously described TALEN pair targeting the *AAVS1/PPP1R12C* locus (*Sanjana et al., 2012*) (hAAVS1 1L TALEN, Addgene plasmid #35431; hAAVS1 1R TALEN, Addgene plasmid #35432). Transfection of these plasmids was performed using FugeneHD (Promega) at a 1:1:8 ratio of L-TALEN:R-TALEN:Donor as previously described (*Lee et al., 2016*). 48 hr after transfection, cells were selected with hygromycin (500 μg/ml) for at least 10 days prior to introducing sgRNAs for CRISPRi-mediated knockdown.

## CRISPRi-mediated knockdown

Single guide RNAs (sgRNAs) targeting a sequence upstream of the endogenous *NORAD* transcription start site or targeting RBMX were cloned into a pU6-sgRNA EF1a-PuroR-T2A-BFP vector (Addgene plasmid #60955). sgRNA sequences are provided in *Supplementary file 1*. pU6-sgRNA vectors were then packaged into lentivirus by transfecting HEK293T cells using a 4:2:1 ratio of pU6-sgRNA: psPAX2:pMD2.G with FuGENE HD. Medium was changed the next day. Media containing the virus was collected and filtered at 48 hr and 72 hr after transfection. Virus was then diluted 1:3 with fresh media and used to transduce HCT116 CRISPRi cell lines overnight in a final polybrene concentration of 8 μg/ml. 48 hr after transduction, selection with 1 μg/ml puromycin was initiated. For HCT116 CRISPRi cells with *AAVS1/NORAD* construct insertion and sgRNA expression, cells were grown in 1 μg/ml puromycin and 500 μg/ml hygromycin.

## UV crosslinking and RNA immunoprecipitation (RIP)

PUM1, PUM2, and RBMX RIP experiments were performed in HCT116 CRISPRi cells stably expressing *AAVS1/NORAD* constructs and depleted of endogenous *NORAD* with CRISPRi as described above. $20 \times 10^6$ cells were washed in cold PBS and UV crosslinked on ice in a Spectrolinker XL-1500 (Spectronics) at 254 nm (400 mJ/cm$^2$). Cells were then scraped, centrifuged, snap-frozen in liquid nitrogen, and stored at −80°C. RIP was performed following a modified eCLIP protocol (*Van Nostrand et al., 2016*) as follows: Cells were lysed in 1 mL cold iCLIP lysis buffer (50 mM Tris-HCl, 100 mM NaCl, 1% NP-40, 0.1% SDS, 0.5% sodium deoxycholate, 1:200 Protease Inhibitor Cocktail III, RNAse inhibitor) for 25 min on ice. Lysed cells were then centrifuged at 14,000 g for 15 min at 4°C and the supernatant was added to pre-washed and antibody-coupled Protein G Dynabeads (Invitrogen). For each RIP, 5 μg of antibody (anti-PUM1, Santa Cruz sc-135049; anti-PUM2, Santa Cruz sc-31535; anti-RBMX, Cell Signaling #14794; Goat IgG control, Santa Cruz sc-2028; Rabbit IgG control Cell Signaling #2729) was coupled to 3.75 mg of beads at room temperature for 45 min, after which unbound antibody was removed. Sample and beads were incubated at 4°C overnight. The next day, beads were washed three times with 900 μL cold High Salt Wash Buffer #1 (50mM Tris-HCl, 1M NaCl, 1 mM EDTA, 1% NP-40, 0.1% SDS, 0.5% sodium deoxycholate) and three times with 500 μL Wash Buffer #2 (20mM Tris-HCl, 10 mM MgCl$_2$, 0.2% Tween-20). Beads were then resuspended in 100 μL Wash Buffer #2, and 70 μL was used for RNA extraction and the remainder for western blotting. Proteins were extracted by incubation in Laemmli buffer for 10 min at 70°C. Antibodies used for western blotting were anti-PUM1 (ab92545, Abcam), anti-PUM2 (ab92390, Abcam), and anti-RBMX (14794, Cell Signaling).

## DNA fluorescence in situ hybridization (DNA FISH)

Aneuploidy in *NORAD* construct rescue experiments was assessed 18 to 21 days after knockdown of endogenous *NORAD*. DNA FISH was performed as described previously (*Kopp et al., 2019*; *Lee et al., 2016*). Chromosome enumeration probes for chromosome 7 (CHR7-10-GR) and chromosome 20 (CHR20-10-RE) were purchased from Empire Genomics. Cells were trypsinized, washed in PBS, and incubated in hypotonic 0.4% KCl solution for 5 min at room temperature. Cells were then fixed in 3:1 methanol:glacial acetic acid and dropped onto slides. DNA FISH hybridizations were performed by the Veripath Cytogenetics laboratory at UT Southwestern. Slides were analyzed using an AxioObserver Z1 microscope (Zeiss). For each sample, 200 nuclei were counted and aneuploidy was defined as a chromosome count that differed from 2 n for at least one of the two tested

chromosomes. Samples were prepared and counted in an experimenter-blinded manner. Two independent HCT116 CRISPRi cell lines stably expressing each *AAVS1* knock-in construct were generated, and each was independently tested for aneuploidy using this method.

## Quantification of chromosome segregation defects

For anaphase nuclei imaging, cells were plated on poly-L-lysine coated coverslips and grown for 24 hr. Samples were then fixed with 4% PFA in PBS for 10 min at room temperature, carefully washed with PBS, rinsed with DEPC-treated water, and mounted using SlowFade Diamond Antifade with DAPI mounting media (Invitrogen). Slides were analyzed using an AxioObserver Z1 microscope. For each sample, 100 anaphase nuclei were imaged and assessed for the presence of chromosome segregation defects. Samples were counted in an experimenter-blinded manner.

## Acknowledgements

We thank Rudolf Jaenisch, Didier Trono, Stanley Qi, Jonathan Weissman, and Feng Zhang for plasmids; Shinichi Nakagawa for technical assistance with RNA FISH; Sushama Sivakumar and Hongtao Yu for assistance with quantification of chromosome segregation defects; Sangeeta Patel in the Veripath Cytogenetics laboratory at UT Southwestern for DNA FISH; and Sungyul Lee, Kathryn O'Donnell, and members of the Mendell laboratory for helpful discussions and comments on the manuscript. This work was supported by grants from CPRIT (RP160249 to JTM), NIH (R35CA197311 to JTM; P30CA142543 to JTM; and P50CA196516 to JTM), and the Welch Foundation (I-1961–20180324 to JTM). JTM is an investigator of the Howard Hughes Medical Institute.

## Additional information

### Funding

| Funder | Grant reference number | Author |
|---|---|---|
| Cancer Prevention and Research Institute of Texas | RP160249 | Joshua T Mendell |
| National Institutes of Health | R35CA197311 | Joshua T Mendell |
| National Institutes of Health | P30CA142543 | Joshua T Mendell |
| National Institutes of Health | P50CA196516 | Joshua T Mendell |
| Welch Foundation | I-1961-20180324 | Joshua T Mendell |
| Howard Hughes Medical Institute | | Joshua T Mendell |

The funders had no role in study design, data collection and interpretation, or the decision to submit the work for publication.

### Author contributions

Mahmoud M Elguindy, Conceptualization, Resources, Data curation, Formal analysis, Investigation, Visualization, Methodology, Writing—original draft, Writing—review and editing; Florian Kopp, Conceptualization, Resources, Data curation, Formal analysis, Investigation, Methodology; Mohammad Goodarzi, Data curation, Software, Formal analysis, Methodology; Frederick Rehfeld, Data curation, Investigation, Methodology; Anu Thomas, Investigation, Methodology; Tsung-Cheng Chang, Resources, Methodology; Joshua T Mendell, Conceptualization, Supervision, Funding acquisition, Visualization, Writing—original draft, Project administration, Writing—review and editing

### Author ORCIDs

Mahmoud M Elguindy  http://orcid.org/0000-0001-9151-1751
Florian Kopp  http://orcid.org/0000-0001-9952-635X
Frederick Rehfeld  http://orcid.org/0000-0002-2751-1025
Joshua T Mendell  https://orcid.org/0000-0001-8479-2284

Decision letter and Author response
Decision letter https://doi.org/10.7554/eLife.48625.017
Author response https://doi.org/10.7554/eLife.48625.018

# Additional files

## Supplementary files

• Supplementary file 1. Sequences of oligonucleotides used in this study.
DOI: https://doi.org/10.7554/eLife.48625.011

• Transparent reporting form
DOI: https://doi.org/10.7554/eLife.48625.012

## Data availability

All data generated or analysed during this study are included in the manuscript and supporting files.

The following previously published dataset was used:

| Author(s) | Year | Dataset title | Dataset URL | Database and Identifier |
|---|---|---|---|---|
| Munschauer M, Nguyen CT, Sirokman K, Hartigan CR, Hogstrom L, Engreitz JM, Ulirsch JC, Fulco CP, Subramanian V, Chen J, Schenone M, Guttman M, Carr SA, Lander ES | 2018 | The NORAD lncRNA assembles a topoisomerase complex critical for DNA replication and genome stability | https://massive.ucsd.edu/ProteoSAFe/dataset.jsp?task=8c85830a749e4bf488e94-f36e8122bbb | MassIVE, MSV0000 82561 |

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
