## [Decision Letter]

Thank you for submitting your article "PUMILIO, but not RBMX, binding is required for regulation of genomic stability by noncoding RNA *NORAD*" for consideration by *eLife*. Your article has been reviewed by two peer reviewers, and the evaluation has been overseen by a Reviewing Editor and James Manley as the Senior Editor. The following individuals involved in review of your submission have agreed to reveal their identity: Igor Ulitsky (Reviewer #1); Philip Grote (Reviewer #2).

The reviewers have discussed the reviews with one another and the Reviewing Editor has drafted this decision to help you prepare a revised submission.

In the manuscript, Mendell and colleagues describe an advance in their study of *NORAD*, a mouse loss-of-function model of which was previously published in *eLife*. In the new manuscript, the authors focus on human cells, where they can reliably test the effects of loss-of-function of *NORAD* an genome stability by measuring fraction of HCT116 cells that become aneuploid, following loss of *NORAD* combined with expression of different rescue constructs. In this setting, the authors test different sequence variants and measure the fraction or aneuploid cells. This system allows the authors to compare the relative contribution of two groups of regions within *NORAD* – the ~20 PUMILIO binding sites, and the region that binds RBMX. RBMX was recently shown to bind *NORAD* by Munschauer et al., which showed that this binding is important for proper replication fork progression, and it was suggested in that paper that this binding is important overall for the roles of *NORAD* in genome stability. The new results convincingly show that PUMILIO binding is important for the aneuploid cell% phenotype (evaluated by measuring two chromosomes with DNA FISH). The authors also convincingly show that *NORAD* is mostly cytoplasmic in unstressed and in DNA-damaged cells, and that in contrast to the previous analysis by Muschauer et al., and consistently with all previous RIP and CLIP data, *NORAD* binds PUM1 and PUM2 PUMILIO proteins.

The topic of *NORAD* function and modes of action is of high interest, and the relative importance of the binding to PUMILIO in the cytoplasm and to RBMX in the nucleus for genome stability, as well as localization of *NORAD*, are a matter of controversy following the conflicting papers. The experiments are generally performed to a high standard and analyzed appropriately. The comments below mainly refer to the way the results are presented in context of prior work from the authors and from Munschauer et al., and are intended to provide a better distinction within the manuscript between the new results and discussion about the results of others.

Essential revisions:

The readout of genome integrity described by Munschauer et al. (% chromosome segregation defects) is not identical to the one described in this submission (and by Lee et al.: "% aneuploid cells") and so formally cannot invalidate Munschauer et al.'s conclusions. Please comment on the possibility that, even with all the new results, *NORAD* has a dual function: as a modulator of PUMILIO factors and as an "assembler" of the NARC1 complex. For a more definitive result, the same phenotypic assays as described in the Munschauer paper would be required. Ideally this should be done, but if the same assay and same readout are not possible then statements (in the Title and Abstract and elsewhere) regarding RBMX binding not being important for genome stability would need to be toned down.

Additional comments:

• The Abstract mentions that much of the prior work published in *eLife* was in mouse, but should also clarify that the new results are mostly in human cells.

• Abstract. What is meant by the data providing "a foundation for further investigation of this pathway"? If not specific then this statement should be removed.

• Introduction section: The first sentence on lncRNAs is too generic and is not true for the majority of lncRNAs – remove/edit.

• "*NORAD* … capacity to bind a large fraction of PUM1/2 within the cell". Is there evidence that more than a few percent of PUM1/2 molecules per cell are bound at any time? Justify. What value is this large fraction?

• "definitively demonstrated that this lncRNA localizes predominantly to the cytoplasm and does not traffic to the nucleus upon induction of DNA damage". The authors do not exclude that some nuclear relocalization does happen. So this sentence (and others, Results paragraph four e.g.) should be toned down.

• Results section paragraph three: The authors should show that these conditions indeed lead to DNA damage.

• Results section: "We conclude that the disparate localization pattern observed using a commercially available RNA FISH probe set (Munschauer et al., 2018) most likely represented a non-specific signal." Speculation, such as this, has a better fit to the Discussion section and here other possibilities should be discussed as well (e.g., differences in the HCT116 clones used in the different labs or differential accessibility of different probe sets to *NORAD* RNPs in the nucleus and in the cytoplasm).

• Results section: "Since these RAP-MS experiments utilized pulse labeling with 4-thiouridine to crosslink *NORAD* to protein interactors, a bias towards detection of proteins that bind to newly synthesized *NORAD* would be expected, likely explaining the enrichment of nuclear interactors observed." This is also speculation and should be moved to the Discussion.

• Subsection “Binding of PUMILIO, but not RBMX, to *NORAD* is necessary for genome stability”: It should be mentioned whether the 5' region the authors use contains any PUMILIO binding sites

• In the same section: What happens to *NORAD* levels in the edited cells (with the extra *NORAD* variants in the AAVS locus) before CRISPRi? Are *NORAD* levels roughly double? This should be mentioned, as these transiently increased levels may affect the phenotype in cells after CRISPRi.

• Subsection “RBMX is not required for *NORAD* expression or localization”: Speculation regarding how RBMX affects *NORAD* levels also should be placed in the Discussion.

• Can the authors please elaborate why the *NORAD*:RBMX study finds PUM only ranking 185th in the RAP-IP experiment and below the significant threshold? Why does applying the strategy with three search engines make this difference?

• There are no biological replicates in Figure 2D; why has Figure 2C biological replicates and not Figure 2D? Also, please provide statistics if the burst of *NORAD* in Figure 2C at 12h is significant.

• Please provide for the 2 independent clones in Figure 4C the analysis as in Figure 3B; does the degree of overexpression differ between the clones?

• Results paragraph one; perhaps "Importantly, a single cell line (human colon cancer cell line HCT116) was used in both of these studies"?

• Results paragraph four – insert "predominantly": "*NORAD* is a predominantly cytoplasmic RNA in HCT116 cells"

• Subsection “PUM1, PUM2, and RBMX are components of the *NORAD* interactome”: Please confirm whether the RAP-MS experiments used HCT116 cells also.

• Subsection “Binding of PUMILIO, but not RBMX, to *NORAD* is necessary for genome stability” paragraph three: *NORAD* "interaction with RBMX was abolished". Is this correct?

• Figure 4C Explain horizontal dotted line and add significance estimates for the different comparisons.

• Figure 4—figure supplement 1. Provide full legend that explains why there are 2 bands in the anti-PUM1 WB and try to improve on the anti-PUM2 WB.

• Subsection “Generation and AAVS1 knock-in of *NORAD* constructs”: typo "and ND4 and were amplified"

---

## [Author Response]

Essential revisions:The readout of genome integrity described by Munschauer et al. (% chromosome segregation defects) is not identical to the one described in this submission (and by Lee et al.: "% aneuploid cells") and so formally cannot invalidate Munschauer et al.'s conclusions. Please comment on the possibility that, even with all the new results, NORAD has a dual function: as a modulator of PUMILIO factors and as an "assembler" of the NARC1 complex. For a more definitive result, the same phenotypic assays as described in the Munschauer paper would be required. Ideally this should be done, but if the same assay and same readout are not possible then statements (in the Title and Abstract and elsewhere) regarding RBMX binding not being important for genome stability would need to be toned down.

We have now performed the identical assay to that performed by Munschauer et al. (quantification of the frequency of chromosome segregation defects), which further demonstrates that RBMX binding is not essential for maintenance of genomic stability by *NORAD*. These new data have been added to Figure 5 of the revised manuscript. Similar to what we reported previously (Lee et al., 2016), and later confirmed by Munschauer et al., *NORAD* knockdown resulted in an increase in the percentage of anaphase cells that display defective chromosome segregation. Expression of full-length *NORAD*, or *NORAD* lacking the RBMX binding site, rescued this phenotype whereas mutation of the PUMILIO binding sites fully abrogated *NORAD* function in this assay.

It is worth noting that we initially chose the DNA FISH assay for the present study because it is a more sensitive readout of genome instability. DNA FISH reports the cumulative effect of chromosome segregation defects over many cell divisions whereas the analysis of mitoses provides a more limited snapshot of the defects present at a single time-point. Nevertheless, we appreciate the reviewers’ suggestion to perform both assays, which have yielded essentially identical results, thereby providing an even more definitive demonstration that PUMILIO binding is essential, whereas RBMX binding is dispensable, for genome maintenance by *NORAD*.

The reviewers have also requested that we comment on the possibility that “*NORAD* has a dual function: as a modulator of PUMILIO factors and as an "assembler" of the NARC1 complex”. First, our data establish that assembly of the NARC1 complex is neither necessary nor sufficient for the genome maintenance function of *NORAD*. Since deletion of the RBMX binding site does not impair *NORAD* function, assembly of NARC1 is not necessary for genome stability. Moreover, the PUMILIO binding site mutant, which retains the ability to interact with RBMX, behaves as a null in genome stability assays, thus demonstrating that assembly of NARC1 is not sufficient for genome maintenance. In light of these observations, the proposed ability of *NORAD* to assemble the NARC1 complex is not relevant to *NORAD*’s genome maintenance function and therefore we chose not to investigate it deeply in this study. Nevertheless, we have performed a limited number of experiments to examine whether *NORAD* is indeed necessary for the assembly of NARC1, as determined by assessing the interaction of RBMX with TOP1 (the readout of NARC1 assembly described in Munschauer et al.). As shown below, in co-immunoprecipitation assays in *NORAD* wild-type and knockout cells performed using epitope-tagged RBMX (using the conditions described in Munschauer et al.), we detect no difference in RBMX:TOP1 interaction in the absence of *NORAD*. We would prefer not to add these data to our manuscript, however, because this experiment is tangential to our main conclusion, which is that NARC1, whether or not it forms, has no role in the genome maintenance function of *NORAD*. Clearly, substantial additional work would be necessary to establish the conditions, if any, under which *NORAD* stimulates assembly of this complex and its functional relevance.

**Author response image 1. respfig1:** Co-immunoprecipitation of RBMX and TOP1 in *NORAD*wild-type and knockout cells. FLAG-tagged RBMX was stably expressed in HCT116 cells and immunoprecipitated using identical conditions as that described in Munschauer et al., 2018.

Additional comments:• The Abstract mentions that much of the prior work published in eLife was in mouse, but should also clarify that the new results are mostly in human cells.

We have edited the Abstract to indicate that the present study utilized human cells.

• Abstract. What is meant by the data providing "a foundation for further investigation of this pathway"? If not specific then this statement should be removed.

By clarifying the mechanism of action of *NORAD*, the data reported in this study provide an “important foundation for further mechanistic dissection of the *NORAD*-PUMILIO axis in genome maintenance”, as it is now phrased in the Abstract.

• Introduction section: The first sentence on lncRNAs is too generic and is not true for the majority of lncRNAs – remove/edit.

The first sentence of the Introduction has been edited to: “Long noncoding RNAs (lncRNAs) have emerged as regulators of diverse biological processes.” This no longer infers that this is true for the majority of lncRNAs, simply that some act as *bona fide* regulators.

• "NORAD … capacity to bind a large fraction of PUM1/2 within the cell". Is there evidence that more than a few percent of PUM1/2 molecules per cell are bound at any time? Justify. What value is this large fraction?

Given our previous measurements of *NORAD* copy number per cell, the number of PUM binding sites per *NORAD* molecule, and the number of PUM1/PUM2 proteins per cell (Lee et al., 2016), it is accurate to state that *NORAD* “has the capacity to bind a large fraction of PUM1/2 within the cell.” As described in that previous publication, there are sufficient PUM binding sites provided by *NORAD* to bind up to 100% of the cellular PUM1/PUM2 pool. However, we agree that it is not clear what fraction of PUM1/2 is actually bound to *NORAD* at any given time, so we have rephrased the sentence to acknowledge that “…it is not yet known whether *NORAD* limits PUM activity through a simple sequestration model or whether additional mechanisms contribute to PUM inhibition.”

• "definitively demonstrated that this lncRNA localizes predominantly to the cytoplasm and does not traffic to the nucleus upon induction of DNA damage". The authors do not exclude that some nuclear relocalization does happen. So this sentence (and others, Results paragraph four e.g.) should be toned down.

We have rephrased the sentence in the Introduction to: “…this lncRNA localizes predominantly to the cytoplasm and does not detectably redistribute to the nucleus upon induction of DNA damage.” Given that the localization pattern of *NORAD* is predominantly cytoplasmic with or without DNA damage, without an increase in nuclear signal under any tested conditions, we believe that this is an accurate description of the results.

Similarly, we revised the relevant sentence in the Results and Discussion section to read: “These comprehensive RNA FISH and subcellular fractionation experiments provide definitive evidence that *NORAD* is a predominantly cytoplasmic RNA in HCT116 cells and does not detectably redistribute to the nucleus upon DNA damage.

• Results section paragraph three: The authors should show that these conditions indeed lead to DNA damage.

We have added γ-H2AX immunofluorescence and western blot data to Figure 2—figure supplement 1 of the revised manuscript, which confirms that doxorubicin and camptothecin treatment lead to DNA damage.

• Results section: "We conclude that the disparate localization pattern observed using a commercially available RNA FISH probe set (Munschauer et al., 2018) most likely represented a non-specific signal." Speculation, such as this, has a better fit to the Discussion section and here other possibilities should be discussed as well (e.g., differences in the HCT116 clones used in the different labs or differential accessibility of different probe sets to NORAD RNPs in the nucleus and in the cytoplasm).

*eLife* allows a versatile format for Research Advance articles and we have chosen to write our manuscript with a combined Results and Discussion section. We believe that this is the most effective means to communicate the results and conclusions of this study. In light of our use of this format, reasonable speculation to provide plausible explanations for disparate results is appropriately located immediately following the presentation of each main experiment. We have edited the text as follows to make it clear that the specific sentence to which the reviewer refers is speculation:

“We speculate that the disparate localization pattern observed using a commercially-available RNA FISH probe set (Munschauer et al., 2018) most likely represented a non-specific signal.”

We do not believe that differences in HCT116 clones provides a likely explanation for the disparate localization patterns observed. The HCT116 cells used in our current and previous localization studies were purchased from ATCC and do not represent a laboratory-specific clone. We and others have reported an identical, cytoplasmic localization pattern in U2OS cells and MEFs (Tichon et al., 2016; Kopp et al., 2019). Moreover, fractionation studies in a large panel of cell lines, including the experiments reported here, have similarly demonstrated cytoplasmic localization of *NORAD* with or without DNA damage (Lee et al., 2016; Tichon et al., 2016; Kopp et al., 2019).

Similarly, we do not believe that differential accessibility of different probe sets to *NORAD* RNPs in the nucleus or cytoplasm provides a plausible explanation for the disparate localization patterns observed. We and others have previously reported predominantly cytoplasmic localization of *NORAD* using RNA FISH with tiled oligonucleotide probe sets (Lee et al.,2016; Tichon et al.,2016), which is the same type of probe used by Munschauer et al. Our new data generated using in vitro transcribed probes spanning the entire length of *NORAD* further support cytoplasmic localization. Equally importantly, the extensive subcellular fractionation studies reported here and elsewhere (Lee et al., 2016; Tichon et al., 2016; Kopp et al., 2019) further demonstrate cytoplasmic localization of *NORAD* with or without DNA damage, providing a completely independent experimental approach to support this conclusion.

• Results section: "Since these RAP-MS experiments utilized pulse labeling with 4-thiouridine to crosslink NORAD to protein interactors, a bias towards detection of proteins that bind to newly synthesized NORAD would be expected, likely explaining the enrichment of nuclear interactors observed." This is also speculation and should be moved to the Discussion.

As described above, this manuscript is formatted with a combined Results and Discussion section. As such, reasonable speculation and data interpretation follows the presentation of each main experiment. Regarding the specific sentence to which the reviewer refers, we have further edited the text to make it clear that we are speculating here:

“Since these RAP-MS experiments utilized pulse labeling with 4-thiouridine to crosslink *NORAD* to protein interactors, a bias towards detection of proteins that bind to newly synthesized *NORAD* would be expected, potentially explaining the enrichment of nuclear interactors observed.”

• Subsection “Binding of PUMILIO, but not RBMX, to NORAD is necessary for genome stability”: It should be mentioned whether the 5' region the authors use contains any PUMILIO binding sites

This fragment does contain one PRE and we have modified the text accordingly:

“An RBMX binding site fragment, representing the 5' end of *NORAD* (nt 33-898), which also harbors one PRE, was also generated (5' fragment).”

A PRE in this segment of *NORAD* is also apparent in Figure 3A.

• In the same section: What happens to NORAD levels in the edited cells (with the extra NORAD variants in the AAVS locus) before CRISPRi? Are NORAD levels roughly double? This should be mentioned, as these transiently increased levels may affect the phenotype in cells after CRISPRi.

These data are provided in Figure 3C and we now explicitly describe these results in the text as follows:

“As expected, in cells infected with a non-targeting control sgRNA (sgControl), increased total levels of *NORAD* were observed upon expression of *NORAD* rescue constructs in *trans* (~1.5-6 fold overexpression, depending on the construct) (Figure 3C). Upon silencing of the endogenous *NORAD* locus, however, near physiologic expression levels for all constructs were achieved, with the exception of the 5' fragment which exhibited ~3-fold overexpression, perhaps indicating increased stability of this transcript segment when expressed in isolation.”

Moreover, we showed that increased *NORAD* expression in control cells does not result in genomic instability. These data were included in our original manuscript and are now shown in Figure 5B of the current version. In the text, we describe these results as follows:

“Importantly, no significant increase in aneuploidy was observed in any of the *NORAD* rescue populations after infection with non-target sgRNA (sgControl), indicating that overexpression of the *NORAD* constructs in *trans* did not trigger genomic instability.”

• Subsection “RBMX is not required for NORAD expression or localization”: Speculation regarding how RBMX affects NORAD levels also should be placed in the Discussion.

As described above, this manuscript is formatted with a combined Results and Discussion section. As such, reasonable speculation and interpretation follows the presentation of each main experiment.

• Can the authors please elaborate why the NORAD:RBMX study finds PUM only ranking 185th in the RAP-IP experiment and below the significant threshold? Why does applying the strategy with three search engines make this difference?

Because the Munschauer et al. study analyzed the RAP-MS experiment using proprietary software that we do not have access to (Spectrum Mill v6.01 pre-release from Agilent Technologies), we are unable to reproduce their analysis and therefore we cannot state definitively why PUM proteins were not more highly ranked in their reported results. However, several aspects of their analysis are notable and may have contributed to their findings. First, we detect two isoforms for both PUM1 and PUM2. We do not know which isoform they refer to as PUM2 (ranked 185 in their analysis) and they do not comment on the PUM1 ranking (which is significantly higher than PUM2 in our analysis). Second, their ranking is based on fold increase in *NORAD* signal compared to signal observed in RMRP pull-down. We do not think RMRP pull-down is the ideal control since different proteins will exhibit different levels of binding to RMRP which may or may not represent background. Instead, we believe that *NORAD* knockout would have been a better control. Regardless, they chose to set a log_2_ fold-increase of 1.6 (*NORAD*/RMRP pull-down), with p < 0.05, as their arbitrary significance threshold. Notably, we observe PUM1 immediately below the enrichment cut-off of log_2_ 1.6, with a more significant p-value than RBMX (see Figure 3—figure supplement 1). Nevertheless, we would prefer not to comment further on their analysis since the data are clear that i) PUM1 and PUM2 are detectable interactors by RAP-MS and ii) these interactions have been confirmed in multiple independent CLIP datasets from human and mouse cells.

• There are no biological replicates in Figure 2D; why has Figure 2C biological replicates and not Figure 2D? Also, please provide statistics if the burst of NORAD in Figure 2C at 12h is significant.

We now provide biological triplicates for the experiments in Figure 2D and Figure 2—figure supplement 1D (subcellular fractionation after treatment with doxorubicin/camptothecin plus actinomycin D). In addition, we provide statistics in Figure 2C and Figure 2—figure supplement 1C, showing that the burst of *NORAD* expression after doxorubicin/camptothecin treatment is indeed statistically significant.

• Please provide for the 2 independent clones in Figure 4C the analysis as in Figure 3B; does the degree of overexpression differ between the clones?

The expression levels of all *NORAD* constructs in both replicates is now provided in Figure 3C of the revised manuscript. Both clones exhibit very similar levels of expression of *NORAD* rescue constructs.

• Results paragraph one; perhaps "Importantly, a single cell line (human colon cancer cell line HCT116) was used in both of these studies"?

We have edited this sentence as follows:

“Importantly, a single cell line (human colon cancer cell line HCT116) was used in both the previous (Lee et al., 2016) and more recent studies (Munschauer et al., 2018), arguing against a cell-type specific difference in *NORAD* trafficking as the cause of these discordant results.”

• Results paragraph four – insert "predominantly": "NORAD is a predominantly cytoplasmic RNA in HCT116 cells"

Edited as suggested.

• Subsection “PUM1, PUM2, and RBMX are components of the NORAD interactome”: Please confirm whether the RAP-MS experiments used HCT116 cells also.

HCT116 cells were indeed used in the RAP-MS experiments and this is now indicated in the text.

• Subsection “Binding of PUMILIO, but not RBMX, to NORAD is necessary for genome stability” paragraph three: NORAD "interaction with RBMX was abolished". Is this correct?

As shown in Figure 4A of the revised manuscript, the 5' deletion and ND4 fragments of *NORAD* exhibit minimal enrichment after RBMX pull-down relative to IgG (~1.0-1.3 fold enrichment). This magnitude of enrichment is indistinguishable from experimental noise. To more accurately describe these results, we revised the text to state that “interaction with RBMX was not detectable above background.”

• Figure 4C Explain horizontal dotted line and add significance estimates for the different comparisons.

The horizontal line indicates the highest level of aneuploidy observed in the sgControl samples. Because this is a complex figure with many data points, we feel that inclusion of this line aids the reader in distinguishing samples with elevated aneuploidy from those without. We have added a similar line to the new Figure 5D (% chromosome segregation defects). Both lines are defined in the figure legend. In addition, significance estimates are provided for all genome stability assays (Figure 5B,D).

• Figure 4—figure supplement 1. Provide full legend that explains why there are 2 bands in the anti-PUM1 WB and try to improve on the anti-PUM2 WB.

We have added a legend to Figure 4—figure supplement 1 that explains that PUM1 migrates as a doublet in HCT116 cells, possibly indicating post-translational modification or alternative splicing. We have confirmed that both bands represent PUM1 protein using siRNA knockdown experiments (not shown). We have replaced the PUM2 and RBMX blots in this figure with higher quality western blots.

• Subsection “Generation and AAVS1 knock-in of NORAD constructs”: typo "and ND4 and were amplified"

Thank you for highlighting this typo, which we have now corrected.